# A Booster with a Genotype-Matched Inactivated Newcastle Disease Virus (NDV) Vaccine Candidate Provides Better Protection against a Virulent Genotype XIII.2 Virus

**DOI:** 10.3390/vaccines11051005

**Published:** 2023-05-20

**Authors:** Ismail Hossain, Jannatul Ferdous Subarna, Congriev Kumar Kabiraj, Jahan Ara Begum, Rokshana Parvin, Mathias Martins, Diego G. Diel, Emdadul Haque Chowdhury, Mohammad Rafiqul Islam, Mohammed Nooruzzaman

**Affiliations:** 1Department of Pathology, Faculty of Veterinary Science, Bangladesh Agricultural University, Mymensingh 2202, Bangladesh; ismail.vpath@bau.edu.bd (I.H.); jf.subarna41286@gmail.com (J.F.S.); congriev@bau.edu.bd (C.K.K.); jahan.begum@bau.edu.bd (J.A.B.); rokshana.parvin@bau.edu.bd (R.P.); emdad001@yahoo.com (E.H.C.); mrislam_bau@yahoo.com (M.R.I.); 2Department of Population Medicine and Diagnostic Sciences, College of Veterinary Medicine, Cornell University, Ithaca, NY 14853, USA; mm3245@cornell.edu (M.M.); dgdiel@cornell.edu (D.G.D.); 3Texas A & M Veterinary Medical Diagnostic Laboratory, 483 Agronomy Rd., College Station, TX 77843-4471, USA

**Keywords:** Newcastle disease, genotype-matched vaccine, genotype XIII.2

## Abstract

Newcastle disease (ND) is endemic in Bangladesh. Locally produced or imported live Newcastle disease virus (NDV) vaccines based on lentogenic virus strains, locally produced live vaccines of the mesogenic Mukteswar strain, as well as imported inactivated vaccines of lentogenic strains, are being used in Bangladesh under different vaccination regimens. Despite these vaccinations, frequent outbreaks of ND are being reported in Bangladesh. Here we compared the efficacy of booster immunization with three different vaccines in chickens that had been primed with two doses of live LaSota vaccine. A total of 30 birds (Group A) were primed with two doses of live LaSota virus (genotype II) vaccine at days 7 and 28, while 20 birds (Group B) remained unvaccinated. At day 60, birds of Group A were divided into three sub-groups, which received booster immunizations with three different vaccines; A1: live LaSota vaccine, A2: inactivated LaSota vaccine, and A3: inactivated genotype XIII.2 vaccine (BD-C161/2010 strain from Bangladesh). Two weeks after booster vaccination (at day 74), all vaccinated birds (A1–A3) and half of the unvaccinated birds (B1) were challenged with a genotype XIII.2 virulent NDV (BD-C161/2010). A moderate antibody response was observed after the primary vaccination, which substantially increased after the booster vaccination in all groups. The mean HI titers induced by the inactivated LaSota vaccine (8.0 log_2_/5.0 log_2_ with LaSota/BD-C161/2010 HI antigen) and the inactivated BD-C161/2010 vaccine (6.7 log_2_/6.2 log_2_ with LaSota/BD-C161/2010 HI antigen) were significantly higher than those induced by the LaSota live booster vaccine (3.6 log_2_/2.6 log_2_ with LaSota/BD-C161/2010 HI antigen). Despite the differences in the antibody titers, all chickens (A1–A3) survived the virulent NDV challenge, while all the unvaccinated challenged birds died. Among the vaccinated groups, however, 50% of the chickens in Group A1 (live LaSota booster immunization) shed virus at 5- and 7-days post challenge (dpc), while 20% and 10% of the chickens in Group A2 (inactivated LaSota booster immunization) shed virus at 3 and 5 dpc, respectively, and only one chicken (10%) in Group A3 shed virus at 5 dpc. In conclusion, the genotype-matched inactivated NDV booster vaccine offers complete clinical protection and a significant reduction in virus shedding.

## 1. Introduction

Newcastle disease (ND) is one of the most important viral diseases in poultry. The virus can infect over 240 bird species and spill over through direct contact between infected and healthy birds [1]. Vaccination against ND is routinely practiced in commercial poultry farming across the world. Despite widespread vaccination, commercial poultry farms in Bangladesh often encounter ND [2,3,4,5,6,7,8,9,10].

ND is caused by avian paramyxovirus-1, or Newcastle disease virus (NDV), which was recently reclassified as avian orthoavulavirus-1 (AOAV-1) in the order Mononegavirales, family Paramyxoviridae, sub-family Avulavirinae, and genus *Orthoavulavirus* [11]. NDV is an enveloped virus containing a non-segmented, single-stranded, negative-sense RNA genome of 15.2 kb that encodes six structural proteins. Among them, the hemagglutinin-neuraminidase (HN) and fusion (F) proteins contribute to virus virulence and immune response [12,13,14,15,16,17]. The pathogenicity of NDV strains is assessed based on standard pathogenicity testing, such as the intracerebral pathogenicity index in day-old chicks and the mean embryo death time in embryonated chicken eggs [18]. NDV strains vary greatly in their pathogenicity and can be divided into three pathotypes: the lentogenic pathotype of mild virulence, the mesogenic pathotype of moderate virulence, and the velogenic pathotype of high virulence [19]. Velogens can be either viscerotropic or neurotropic based on their tissue tropism in infected birds [20].

The genotypic classification of NDVs relies on the F gene sequences. Two classes of NDVs, class I (one genotype) and class II (21 genotypes, I–XXI), have been recognized. Class I viruses are mostly apathogenic for chickens and are detected in various waterfowl, shorebirds, and birds from live bird markets [21,22,23,24]. Class II contains both avirulent and virulent strains isolated from both domestic poultry and wild birds [25,26]. In Bangladesh, NDVs belonging to genotype XIII.2 are circulating in chickens and genotype XXI.1.2 in pigeons [2,8,10,27]. NDVs of genotype XIII have also been reported in India, Pakistan, Indonesia, Iran, Sweden, Russia, Kazakhstan, Burundi, Tanzania, Zambia, and South Africa [22,26,28,29,30,31,32]. Very recently, NDV belonging to genotype VII.2 has been reported in chickens in Bangladesh [9].

NDV isolates belong to a single serotype. The classical NDV vaccines are capable of preventing clinical disease; however, they fail to prevent virus shedding upon challenge with heterologous field strains [33,34,35]. While antigenic matches at F and HN proteins between the vaccine strain and challenge virus significantly improve clinical protection and reduce challenge virus shedding [34,36]. Several studies demonstrated the relative advantages of genotype-matched NDV vaccines in preventing challenge virus shedding against genotypes V, VII, and XII, making the genotype-matched vaccine an important option in controlling ND globally [34,35,37,38,39,40]. Beside genotypic mismatch, poor vaccination practices, immunosuppression and co-infection could also lead to the immune escape of field viruses from the vaccine [41].

Vaccination against ND in Bangladesh is widely practiced in commercial poultry and breeder flocks [42,43,44]. Primary vaccination is performed with a live vaccine based on a lentogenic strain (LaSota, B1, or F strain), while for booster vaccination, either a lentogenic live vaccine, a lentogenic killed vaccine, or, less frequently, a mesogenic live vaccine (Mukteswar strain) is used. Still, a high prevalence of ND outbreaks is being reported in Bangladesh [2,3,4,5,6,7,8,9,10,45]. Among different factors, genotypic mismatches between the circulating viruses and vaccine strains could potentially contribute to vaccine failure in the field [33,35]. Here, we assessed the immunogenicity and protective efficacy of three different vaccination regimens against a velogenic genotype XIII.2 NDV challenge, where the primary vaccination was done with a live vaccine of the lentogenic strain, and the booster vaccination was done with either a live or inactivated vaccine of the lentogenic stain or an inactivated vaccine prepared with a velogenic local strain genetically matching the challenge virus. Our study showed that booster with genotype-matched inactivated vaccine gives better protection with significantly reduced virus shedding against a velogenic NDV challenge.

## 2. Materials and Methods

### 2.1. Virus and Vaccine

Two NDV strains belonging to two different genotypes, the LaSota (genotype II) vaccine strain (PoulShot^®^ LaSota, CAVAC, Daejeon, Korea (marketed by Pharma and Firm, Dhaka, Bangladesh)) and a Bangladeshi field isolate BD-C161/2010 (genotype XIII.2), were used in this study. The LaSota virus was isolated from the commercial live LaSota vaccine. For this, the lyophilized vaccine was reconstituted in phosphate buffered saline (PBS) containing antibiotics (gentamicin, 10 mg/mL) and propagated in embryonated chicken eggs. The reisolated LaSota virus stock had a titer of 10^8.7^ EID_50_/mL. BD-C161/2010 is a local virulent strain of NDV that has been isolated from a natural outbreak in chicken [6]. The field strain has been classified as genotype XIII.2 [8] and belongs to the velogenic viscerotropic pathotype [46]. The stock of BD-C161/2010 had a titer of 10^8.3^ EID_50_/mL.

### 2.2. Virus Inactivation and Preparation of NDV Inactivated Vaccines

The allantoic fluid containing two NDV isolates (LaSota and BD-C161/2010) was diluted in PBS to have a titer of 10^8^ EID_50_ per ml and inactivated with 0.4% formalin at room temperature for 2 h. To check the virus inactivation, 200 µL of inactivated virus was inoculated into three 9-day-old embryonated chicken eggs, and the embryo mortality was monitored regularly for 6 days for three blind passages. The absence of embryo mortality and no hemagglutinating activity in the allantoic fluid indicated complete inactivation of the virus. For the preparation of the vaccine with the inactivated LaSota and BD-C161/2010 viruses, the allantoic fluid was emulsified with an equal volume of incomplete Freund’s adjuvant (IFA) (Sigma–Aldrich, St. Louis, MO, USA) using a mortar and pestle. For the live virus vaccine of the LaSota strain, the allantoic fluid was used directly without further processing.

### 2.3. Vaccination and Challenge

We used a vaccination regimen commonly practiced on commercial poultry farms in Bangladesh. A total of 50 ISA Brown chicks were obtained from a commercial company. The chicks were raised following strict biosecurity protocols. On day 7, chicks were allocated into two groups: group A (*n* = 30) and group B (*n* = 20). Birds in Group A received 100 µL of live vaccine of the LaSota virus strain (10^8^ EID_50_/mL) using both intranasal (i/n) and intraocular (i/o) routes at days 7 and 28. While Group B birds remained as unvaccinated controls (sham immunized) and received 100 µL of PBS through the same routes. At day 60, birds from Group A were subdivided into three groups (A1, A2, and A3, *n* = 10 each) and received boosters with three different vaccines: Group A1 birds received live LaSota booster (100 µL, 10^8^ EID_50_/mL, i/n, i/o) and are hereafter designated as LaSota (Live); Group A2 birds received 500 µL of inactivated LaSota vaccine subcutaneously (hereafter designated as LaSota (Killed)); and Group A3 birds received 500 µL of inactivated BD-C161/2010 vaccine subcutaneously (hereafter designated as BD-C161 (Killed)).

Two weeks after the booster vaccination (day 74), the three vaccinated groups (A1, A2, and A3) were challenged with 200 µL (10^6^ EID_50_/mL) of BD-C161/2010 (genotype XIII.2) via intranasal and intraocular routes (100 µL per route). The birds of the unvaccinated Group B were allocated into two subgroups (B1 and B2), with 10 birds in each group. The birds of B1 were inoculated with the BD-C161/2010 virus as above (unvaccinated challenged group), while the Group B2 received 200 µL of sterile PBS via the same routes (unvaccinated unchallenged group). Clinical signs and mortality were recorded twice daily for 14 days post-challenge (dpc). The experimental layout is presented in Figure 1.

### 2.4. Sample Collection and Processing

To check the level of maternally derived antibody (MDA), blood samples (*n* = 12) were collected randomly from six chicks each from groups A and B at day 7. In addition, blood samples were collected from each bird in each sub-group on days 28, 42, 60, 74, and 88. Sera were separated and centrifuged at 3000 rpm for 10 min, inactivated at 56 °C for 30 min, and stored at −20 °C. For virus shedding analysis, oropharyngeal and cloacal swabs were collected from all birds at 5- and 7-day post-challenge (dpc) in 1 mL of sterile PBS containing gentamicin (10 mg/mL) and stored at −20 °C. If the birds died earlier, the swabs were collected during the necropsy.

### 2.5. Hemagglutination Inhibition Test

The serum antibody level of birds was determined by a standard hemagglutination inhibition (HI) test using 4 HA units (HAU) of LaSota and BD-C161/2010 antigen, following the protocols [18]. Briefly, 25 µL of PBS was dispensed into each well of a V-bottomed microtiter plate. A total of 25 µL of heat-inactivated serum was added to the first well, and a serial two-fold dilution was made across the plate. In addition, 25 µL of 4 HAU of virus antigen was added to each well and incubated for 30 min at room temperature. Further, 25 µL of 1% chicken RBC was added to each well, mixed gently, and incubated for 40 min at room temperature. The agglutination was assessed by tilting the plate. The HI titer was calculated as the highest dilution of serum causing complete inhibition of 4 HAU of antigen [18].

### 2.6. Detection of Virus Shedding by RT-qPCR

The shedding of virus in the oropharyngeal swabs and cloacal swabs following challenge was determined by RT-qPCR. Viral RNA was extracted from swab samples using a commercial kit (Purelink^TM^ Viral RNA/DNA Mini kit, ThermoFisher Scientific, Waltham, MA, USA). RNA quantity was measured using a Nanodrop spectrophotometer (Nanodrop One, ThermoFisher Scientific, Waltham, MA, USA). The viral RNA load in the swab samples was quantified by RT-qPCR targeting the nucleoprotein gene using the Luna^®^ universal one-step RT-PCR kit (New England Biolabs, Ipswich, MA, USA) as described previously [10,45]. Samples with Ct (cycle threshold) values of ≤37 in both replicates were considered positive.

### 2.7. Sequencing

To check if the NDV detected in the oropharyngeal and cloacal swabs was the vaccine virus or the challenge virus, a fragment of the F gene was amplified and sequenced. We selected five oropharyngeal and six cloacal swab samples from the LaSota (live) booster-vaccinated chickens (Group A1) for sequencing. A 953 bp fragment from the 5′ end of the F gene was amplified using primer pairs NDVF13-F1 5′-GAC GCA ACA TGG GCT CCA RAY CTT-3′, NDVF13-R1 5′-GGC AAA CCC TCT GGT CGT GCT YAC-3′ as described previously [8]. The SuperScript™ III One-Step RT-PCR System with Platinum™ Taq High Fidelity DNA Polymerase (Invitrogen, Waltham, MA, USA) was used following the manufacturer’s instructions. The RT-PCR amplicons were purified using the FavorPrep™ GEL/PCR Purification Kit (Favorgen Biotech Corp., Ping Tung, Taiwan) and sequenced commercially (1st BASE, Selangor, Malaysia). The sequences were edited and analyzed using Bioedit and MEGA7 (www.megasoftware.net) software. Further, the complete F gene sequence of the LaSota vaccine strain was also obtained, as described previously [8]. The complete F gene sequence of BD-C161/2010 was retrieved from GenBank (Accession number MK934289.3).

### 2.8. Statistical Analysis

To compare the antibody responses between the vaccinated groups, a two-way ANOVA with a Bonferroni post hoc test was applied. A *p*-value of ≤0.05 was considered significant. Analysis was conducted using GraphPad Prism 9 (GraphPad Software Inc., Boston, MA, USA).

## 3. Results

### 3.1. Antibody Response following Vaccination and Challenge

All chicks had a relatively high maternal antibody (MDA) level as detected at day 7 (mean HI titer 5.3 log_2_), which gradually dropped to 2.16 and 0.4 log_2_ at days 28 and 42, respectively, and completely disappeared by day 60 in the unvaccinated chickens (Figure 2a). However, following two doses of primary vaccination with LaSota live vaccine at days 7 and 28, the antibody level moderately increased to a mean titer of 4.0 log_2_ but declined to a mean titer of 2.7 log_2_ at day 60 (Figure 2a). The antibody levels remarkably increased after booster vaccination at day 60; however, the antibody levels varied with the type of booster vaccine used (Figure 2b). The HI titer also varied depending on the type of antigen used in the test. Following booster vaccination, both inactivated vaccines (genotype II LaSota strain and genotype XIII.2 BD-C161/2010 strain) induced significantly higher antibody titers as compared to those induced by immunization with the live LaSota vaccine. Fourteen days after booster vaccination with live LaSota vaccine (Group A1), the mean HI titer was relatively low, 3.6 log_2_ (with LaSota HI antigen) or 2.6 log_2_ (with BD-C161/2010 HI antigen). On the contrary, after immunization with the inactivated LaSota vaccine, the mean HI titer was 8.0 log_2_ (with LaSota HI antigen) or 5.0 log_2_ (with BD-C161/2010 HI antigen). Similarly, after booster vaccination with the inactivated BD-C161/2010 vaccine, the mean HI titer was 6.7 log_2_ (with LaSota HI antigen) or 6.2 log_2_ (with BD-C161/2010 HI antigen). The antibody titers further increased upon challenge in all three vaccinated groups, irrespective of the type of booster vaccine used, with no significant differences between the three groups at day 88 (14 days post-challenge) (Figure 2c).

### 3.2. Clinical Signs, Mortality, and Postmortem Findings following Challenge

Following challenge infection, all the unvaccinated challenged chickens (Group B1) developed typical signs of velogenic viscerotropic NDV infection. The common clinical signs were severe depression with respiratory distress (10/10), diarrhea (7/10), ocular discharges (4/10), nervous signs of torticollis (3/10), and nasal discharges (2/10) (Table 1). Among the vaccinated groups, only three chickens in Group A1, which received the live LaSota vaccine for both primary and booster vaccinations, showed diarrhea. While the remaining chickens of Group A1 and all chickens of Groups A2 and A3 received either inactivated LaSota vaccine or inactivated BD-C161/2010 vaccine (Group A3) as boosters, respectively, they remained healthy, similar to the unvaccinated, unchallenged control birds of Group B2 (Table 1). All 10 birds in the unvaccinated challenged group (A1) died between 3- and 5-days post challenge (dpc), while all birds in the three vaccinated groups survived (Figure 3).

At necropsy, the unvaccinated challenged birds (Group B1) showed lesions typical of velogenic viscerotropic NDV infection, including severe congestion in the lungs, hemorrhages in the trachea, necrotic foci on the spleen, congestion in the brain, hemorrhages on the tip of the proventricular glands, and button-like ulcers in the intestine (Figure 4a–d). Other than a slight congestion in the lungs of some birds, no such pathological changes were observed in any of the vaccinated chickens (Figure 4e–h) or the control uninfected chickens, which were euthanized at the end of the experiment at 14 dpc.

### 3.3. Virus Shedding following Challenge

Virus shedding was assessed in all the challenged birds (Table 2). Oropharyngeal and cloacal swabs were collected from the vaccinated and challenged birds (Groups A1, A2, and A3) at 5 and 7 dpc and from the sham immunized and challenged group (Group B1) at 3, 4, and 5 dpc (at necropsy). The samples were tested by RT-qPCR. All (10/10) oropharyngeal and cloacal swabs collected from the unvaccinated challenged chickens that died between 3 and 5 dpc were positive for NDV RNA, with Ct values ranging from 16.6–25.0 and 18.1–25.0 for oropharyngeal and cloacal swabs, respectively (Table 2).

The vaccinated challenged birds shed virus sporadically, which was most frequent in the birds receiving the LaSota live booster vaccine (Group A1). In this group, 3 oropharyngeal swabs and 5 cloacal swabs (out of 10) were positive in RT-qPCR at 5 dpc with Ct values of 31.5-33.2 and 26.2-32.4, respectively. Similarly, 5 oropharyngeal swabs and 3 cloacal swabs were positive for NDV at 7 dpc with Ct values of 22.3-30.1 and 19.6–23.0, respectively. Of note, the viral RNA load was relatively higher at 7 dpc than 5 dpc in these birds, as indicated by lower Ct values. On the other hand, virus shedding was less frequent in chickens receiving the inactivated booster vaccines (Groups A2 and A3) as compared to the live booster vaccine group (Group A1). In the LaSota inactivated booster vaccine group (A2), oropharyngeal swabs of 2 chickens at 5 dpc and 1 chicken at 7 dpc were positive for NDV. While in the inactivated BD-C161/2010 booster vaccinated group (A3), the oropharyngeal swab of only one chicken was positive at 5 dpc. Collectively, the booster with an inactivated virus vaccine greatly reduced the virus shedding as compared to the live booster vaccine, and the genotype-matched inactivated booster vaccine (BD-C161/2010 (killed)) almost prevented the virus shedding.

To test whether the NDV detected in swab samples was from the LaSota (live) booster vaccine virus or the challenge virus BD-C161/2010, partial genome sequencing was performed. Nucleotide sequences of the first 945 base pairs (315 amino acids) of the fusion (F) protein of NDV from 11 oropharyngeal/cloacal swab samples collected from LaSota live booster vaccinated chickens were compared with the corresponding nucleotide sequences of LaSota and BD-C161/2010 strains. Nucleotide alignment analysis revealed that all the swab samples had NDV RNA with 100% nucleotide sequence similarity to that of the challenge strain BD-C161/2010.

### 3.4. Genetic Variability between LaSota and BD-C161/2010 NDV Strains at Putative Neutralizing Epitopes

A total of five neutralizing epitopes (A1–A5) on the F-protein of NDV have been identified [47]. Among them, four (A1, A2, A3, and A5) were discontinuous epitopes, and one (A4) was a linear epitope. Comparative amino acid residue analysis between the LaSota (genotype II) and BD-C161/2010 (genotype XIII.2) strains revealed conserved amino acid residues at the four antigenic sites at A1(^72^D), A2(^78^K), A5(^343^L) and A4 (^157^SIAATNEAVHEVTDG^171^) epitopes of F-protein (Table 3). However, an A79T mutation at the A3 neutralizing epitope was noticed in the BD-C161/2010 strain.

## 4. Discussion

This study evaluated the immunogenicity and protective efficacy of three different types of NDV booster vaccination in chickens against a virulent challenge. The vaccines used for boosters were a live vaccine of the LaSota strain (genotype II) and two inactivated vaccines of the LaSota strain and the BD-C161/2010 strain (genotype XIII.2). The boosters were applied following initial priming with two doses of live vaccine with the LaSota strain. The challenge virus was a local virulent isolate, BD-C161/2010 (genotype XIII.2). The efficacy of booster vaccines was evaluated in terms of antibody responses, clinical protection, and virus shedding. All three vaccines offered 100% clinical protection against virulent NDV challenges; however, boosters with inactivated vaccines induced a higher antibody response and reduced virus shedding. In particular, the booster vaccination with a genotype-matched inactivated vaccine (where the vaccine virus genotype is similar to the challenge virus) nearly completely prevented virus shedding.

The ND vaccines are intended to (i) decrease or eliminate clinical disease, (ii) reduce virulent virus shedding, and (iii) increase the infectious dose of the challenge virus [48]. Although the efficacy of current field vaccination is evaluated mostly based on the first goal due to technological limitations in the field to assess virus shedding. Both traditional and recombinant vaccines are being used in the field, following different vaccination strategies depending on specific ND epidemiological situations and vaccine availability [41]. Live vaccines are easy to administer using mass application techniques and induce both mucosal and humoral immune responses. A single dose of live vaccine can provide complete clinical protection but does not prevent challenge virus shedding [34,41,49]. Increasing the dose of live vaccine can significantly reduce virus shedding; however, it increases the cost of vaccines [41]. Moreover, in ND-endemic countries, the high MDA in chicks can hinder the immune response to live vaccines in commercial poultry flocks [50]. On the other hand, inactivated vaccines, which can be formulated as genotype-matched vaccines, mount a stronger humoral immune response in birds than live vaccines and provide complete clinical protection, but they do not prevent virus shedding alone [51]. The immunity induced by inactivated vaccines is less affected by the presence of MDA in chicks [52]. Most of the inactivated vaccines are applied as boosters following priming with a live attenuated strain, and this combination of live and inactivated vaccines has been shown to be very effective in reducing challenge virus shedding [53,54].

The study was conducted on commercial chickens with MDA. The MDA level was quite high, and at the time points of primary vaccination (day 7 and day 28), the mean HI titer of residual MDA was 5.3 log_2_ and 2.16 log_2_, as observed in the unvaccinated chickens. Apparently, the high MDA level interfered with the uptake of the primary vaccine, as there was no rise in antibody levels in the vaccinated birds after the first shot as compared to the unvaccinated birds; however, the antibody titer increased two weeks after the second immunization but declined over the next two weeks, leaving the birds vulnerable to infection in high field challenges. This would suggest that the timing of primary vaccination (day 7 and day 28), currently practiced in the country, needs to be reconsidered to induce and maintain sustained vaccinal immunity.

The booster immunization given on day 60 greatly enhanced antibody responses. The results revealed that booster vaccination with an inactivated vaccine prepared either from the LaSota strain (genotype II) or BD-C161/2010 (genotype XIII.2) induced significantly higher antibody responses as compared to the LaSota live booster vaccination. It has been reported earlier that inactivated vaccines could induce higher antibody responses in chickens when compared to live virus vaccines [55]. The variations in HI titer observed with different HI antigens deserve attention. In LaSota (live or inactivated) virus booster vaccine groups, the HI titer was remarkably higher when tested with homologous LaSota HI antigen as compared to heterologous BD-C161/2010 antigen. In the inactivated BD-C161/2010 virus booster vaccine group, the HI titer was almost comparable with homologous BD-C161/2010 and heterologous LaSota HI antigen. Therefore, for proper evaluation of the antibody response by HI testing, it may be wise to use the HI antigen prepared from the likely challenge strain of NDV in the field.

Despite some variations in the magnitude of antibody response, all three vaccination regimens conferred 100% clinical protection against challenge with a velogenic NDV field strain (BD-C161/2010) of genotype XIII.2, though three out of ten birds in Group A1 (live LaSota booster) developed transient diarrhea. On the other hand, all unvaccinated challenged birds (Group B1) developed respiratory distress accompanied by diarrhea, conjunctivitis, and nervous signs and died between days 3 and 5 after the challenge.

Next, we evaluated the level of protection provided against the virulent NDV challenge. Virus shedding results determined by RT-qPCR showed (Table 2) that 50% of birds in Group A1 (LaSota live virus booster vaccine) shed virus at 5 and 7 dpc. On the other hand, in Group A2 (LaSota inactivated booster vaccine), 20% and 10% of birds shed virus at 5 and 7 dpc, respectively, while in Group A3 (BD-C161/2010 inactivated booster vaccine), only a single bird (10%) shed virus at 5 dpc. These results demonstrate that boosters with inactivated vaccine after priming with live vaccine could reduce virus shedding significantly as compared to live vaccine boosters, which may be related to neutralization or faster clearance of the live vaccine virus [40,56]. The findings also suggest that an inactivated vaccine prepared from a strain genotypically matched with the challenge strain may perform better in the prevention of virus shedding.

All Newcastle disease viruses belong to the same serotype, and viruses of different genotypes cross-protect each other. However, the failure to achieve sterile immunity following vaccination against NDV could be due to a suboptimal immune response as well as a genotypic mismatch between the vaccine virus and the challenge virus [41,57]. NDV is evolving constantly, leading to the emergence of new sub/genotypes, which could be a challenge for genotype-matched vaccines. Furthermore, companies that work with virulent NDV require higher laboratory biosecurity, and most countries do not have vaccine companies that have the level of biosecurity to safely handle virulent NDV [41]. However, genotype-matched vaccines showed higher efficacy in decreasing morbidity and mortality rates than genotype-heterologous vaccines, even when birds were suboptimally vaccinated with low doses of vaccines given only seven days before challenge with a virulent NDV [58]. In countries where more than one genotype of virulent NDV is circulating, a multivalent inactivated vaccine could be formulated.

In the present study, a comparison between the gene sequences of the LaSota strain (genotype II) and BD-C161/2020 (genotype XIII.2) revealed an amino acid substitution (A79T) at the putative neutralizing epitope A3 in the fusion protein [47]. Antigenic variations between phylogenetically divergent NDVs have been demonstrated previously [57,59].

## 5. Conclusions

Two shots of primary vaccination with a live virus vaccine followed by booster vaccination with an inactivated vaccine provided a better antibody response and reduced virus shedding following a virulent challenge as compared to both priming and booster vaccination with a live virus vaccine. Further, an inactivated booster vaccine prepared with a strain genotypically matching the challenge virus provides even better protection in terms of the prevention of virus shedding. Therefore, we recommend two priming vaccinations using LaSota or similar live attenuated NDV strains followed by booster vaccination with genotype-matched inactivated vaccine produced from a local field isolate to control the ND outbreaks in poultry in the situation of widespread circulation of velogenic NDV.

## Figures and Tables

**Figure 1 vaccines-11-01005-f001:**
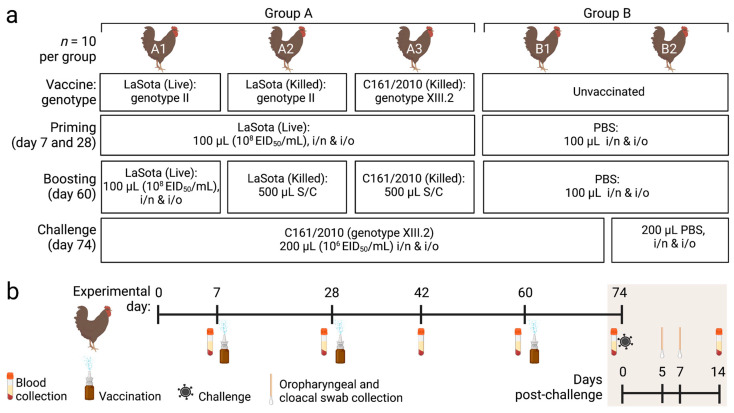
Schematic presentation of the experimental layout. Illustration showing (**a**) the vaccination and challenge scheme of the experiment and (**b**) sample types and collection days. PBS-phosphate buffered saline, i/n: intranasal, i/o: intraocular, S/C: subcutaneous. The illustration was prepared using BioRender (https://www.biorender.com/).

**Figure 2 vaccines-11-01005-f002:**
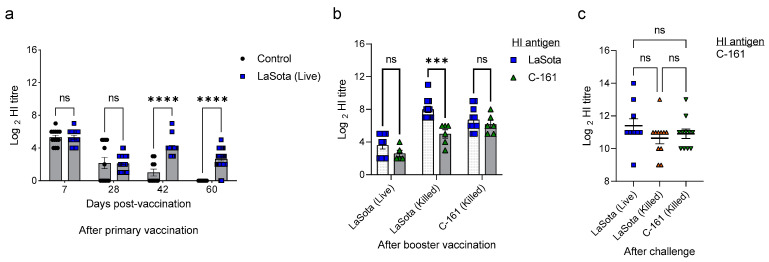
Antibody titers in sera of chickens vaccinated with different formulations of NDV vaccines. (**a**) Dot plot showing HI titers of control and LaSota (live) vaccinated chickens at days 7, 28, 42, and 60. (**b**) Dot plot showing HI titers of booster-vaccinated chickens at day 74 (2 weeks after booster vaccination). (**c**) Dot plot showing HI titers of chickens at day 88 (2 weeks after challenge). Data indicates mean ± SEM. Two-way (**a**,**b**) and one-way (**c**) ANOVA with Bonferroni multiple comparison test, *** *p* ≤ 0.01, **** *p* ≤ 0.0001, ns = not significant.

**Figure 3 vaccines-11-01005-f003:**
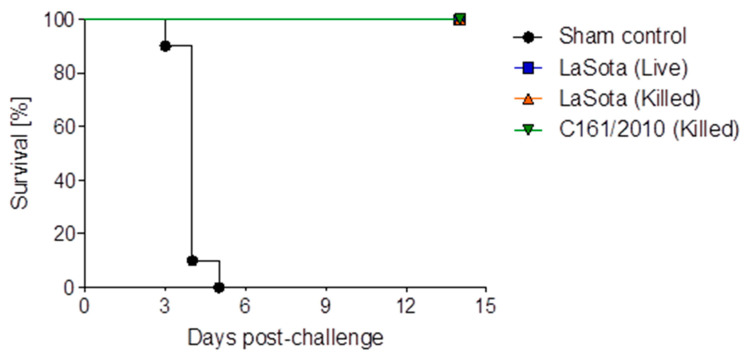
Survival curve of NDV challenged chickens receiving three different booster vaccine formulations.

**Figure 4 vaccines-11-01005-f004:**
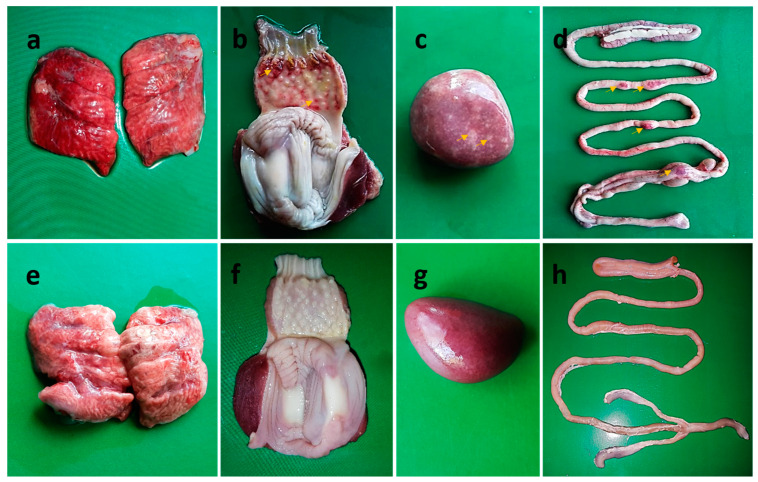
Gross pathological lesions in unvaccinated (**a**–**d**) and BD-C161/2010 booster immunized (**e**–**h**) chickens following challenge with NDV strain BD-C161/2010. Unvaccinated NDV-challenged chickens showing (**a**) marked hemorrhages and congestions in lungs; (**b**) hemorrhages on the tips of proventricular glands (arrows); (**c**) mottling (numerous necrotic white spots) of the spleen (arrows) with atrophy, and (**d**) button-like ulcers (arrows) in the intestine. Chickens from the BD-C161/2010 booster immunized group showing slight congestion in the lungs (**e**) and normal-appearing proventriculus (**f**), spleen (**g**), and intestine (**h**). Chickens receiving both the LaSota (live) and LaSota (killed) booster immunizations had gross lesions similar to the BD-C161/2010 (killed) booster immunized group.

**Table 1 vaccines-11-01005-t001:** Comparative data on clinical signs and mortality of chickens belonging to different experimental groups.

Groups	Booster Vaccination	Challenge	Respiratory Distress	Ocular Discharge	Nasal Discharge	Diarrhea	Nervous Sign
A1	LaSota (Live)	Yes	0/10 *	0/10	0/10	3/10	0/10
A2	LaSota (Killed)	Yes	0/10	0/10	0/10	0/10	0/10
A3	BD-C161/2010 (Killed)	Yes	0/10	0/10	0/10	0/10	0/10
B1	Unvaccinated	Yes	10/10	4/10	2/10	7/10	3/10
B2	Unvaccinated	No	0/10	0/10	0/10	0/10	0/10

Note: * Number of chickens showing the signs/number of chickens under observation.

**Table 2 vaccines-11-01005-t002:** Comparison of virus shedding in oropharyngeal and cloacal swabs of chickens belonging to different experimental groups.

		5 dpc (Group A1, A2, A3)/3-5 dpc (Group B1)				7 dpc			
Group	Booster Vaccine	Oropharyngeal Swab		Cloacal Swab		Oropharyngeal Swab		Cloacal Swab	
		No.	Ct	No.	Ct	No.	Ct	No.	Ct
A1	LaSota (Live)	3/10	31.5–33.2	5/10	26.2–32.4	5/10	22.3–30.1	3/10	19.6–23.0
A2	LaSota (Killed)	2/10	31.4–31.7	0/10	-	1/10	31.8	0/10	-
A3	BD-C161/2010 (Killed)	1/10	32.6	0/10	-	0/10	-	0/10	-
B1	Unvaccinated	10/10	16.7–25.1	10/10	18.1–25.0	N/A	-	N/A	-

Note: dpc—days post-challenge, Ct—Cycle threshold, N/A—Not available.

**Table 3 vaccines-11-01005-t003:** Deduced amino acid residues at the predicted neutralizing epitopes of NDV between the LaSota and BD-C161/2010 strains.

Epitopes	A1	A2	A3	A4															A5
Position	72	78	79	157	158	159	160	161	162	163	164	165	166	167	168	169	170	171	343
LaSota	D	K	A	S	I	A	A	T	N	E	A	V	H	E	V	T	D	G	L
BD-C161/2010	•	•	T	•	•	•	•	•	•	•	•	•	•	•	•	•	•	•	•

Note: Residues matching with the LaSota strain are marked with dots.

## Data Availability

All authors agree that the data presented in this study are openly available through the MDPI publisher platform or others without any restriction. Data supporting the reported results can be available upon request to the corresponding author.

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
