# Peer review of "A Booster with a Genotype-Matched Inactivated Newcastle Disease Virus (NDV) Vaccine Candidate Provides Better Protection against a Virulent Genotype XIII.2 Virus"

_vaccines, 2023, doi:10.3390/vaccines11051005_

Round 1

Reviewer 1 Report

The authors compared the efficacy of booster immunisation of three different vaccines in chickens that had previously received two doses of LaSota vaccine. The study was conducted in a fair and comprehensive manner. The results demonstrated the efficacy of booster immunisation. Have other methods or protocols been used and described experimentally to increase the efficacy of the vaccine immunization in the recent bibliography? It might be useful to compare the various procedures in the discussion.

Author Response

Thank you for the appreciation. The efficacy of various vaccine methods has been discussed (Line 358-376)

Reviewer 2 Report

The authors of submitted manuscript designed their study to evaluate three different type of Newcastle Disease Virus in chicken against the virulent challenge. The authors used the booster  live vaccine of LaSota strain (genotype II) and two inactivated vaccines of LaSota strain and BD-C161/2010 strain (genotype XIII.2). The efficacy of booster vaccines was evaluated in terms of antibody responses, clinical protection, and  virus shedding after they challenged the vaccinated and non vaccinated chickens with the live virus. 

their data show that all the three vaccines protected 100% of the chickens against virulent NDV challenge. 

The study designed looks proper and the methodology written clearly. The authors explained the data clearly in details and discussed their finding very well. I have no comments for the authors to improve their manuscript. 

Author Response

Thank you for your appreciation.

Reviewer 3 Report

Newcastle disease virus (NDV) is endemic and responsible for widespread outbreaks in Bangladesh, as well as other areas across the world.  These outbreaks occur despite the existence of varying vaccination programs.  In Bangladesh, vaccination is commonly based on either one of the common lentogenic strains or the mesogenic Mukteswar strain; while primary vaccination utilizes a live lentogenic strain, booster vaccination is with either a live or killed lentogenic strain or live Mukteswar virus.  Still, outbreaks of Newcastle disease occur frequently. 

This study attempts to improve on the efficacy of vaccination in Bangladesh by comparing the immunogenicity and protective capacity of three different vaccination regimens, in which primary vaccination with a live lentogenic vaccine is followed by a booster with either a live or inactivated lentogenic strain or an inactivated, genotypically-matched local velogenic strain.  Although all three vaccine regimens conferred complete clinical protection against challenge with the velogenic strain, boost with the genotypically-matched velogenic strain was far more effective at preventing virus shedding.  These findings confirm that the use of the local velogenic strain as boost vaccine is a more efficacious strategy to prevent Newcastle disease outbreaks.  In addition, the authors go on to show that, of the five known neutralizing epitopes on the NDV F protein relative to those on the lentogenic LaSota strain, only one exhibits a substitution, that being an A79T mutation the A3 antigenic site.  This could mean that efficacy of the booster vaccine strain depends to an extent on its homology with the challenge virus with a genotypic mismatch between vaccine and challenge virus resulting in a suboptimal protective response.

This a very strong manuscript.  The experiments are all well designed and appropriately interpreted.  The data strongly support the use of a vaccination strategy that involves two priming vaccination with a live lentogenic strain followed by a booster vaccination with a genotypically-matched inactivated local virus.  There is only one minor criticism of the manuscript.  In the interests of completeness, arrows should be added to Figure 4 to indicate, wherever possible, the pathological lesions in panels a-d with an explanation added to the figure legend. 

Author Response

Newcastle disease virus (NDV) is endemic and responsible for widespread outbreaks in Bangladesh, as well as other areas across the world.  These outbreaks occur despite the existence of varying vaccination programs.  In Bangladesh, vaccination is commonly based on either one of the common lentogenic strains or the mesogenic Mukteswar strain; while primary vaccination utilizes a live lentogenic strain, booster vaccination is with either a live or killed lentogenic strain or live Mukteswar virus.  Still, outbreaks of Newcastle disease occur frequently. 

This study attempts to improve on the efficacy of vaccination in Bangladesh by comparing the immunogenicity and protective capacity of three different vaccination regimens, in which primary vaccination with a live lentogenic vaccine is followed by a booster with either a live or inactivated lentogenic strain or an inactivated, genotypically-matched local velogenic strain.  Although all three vaccine regimens conferred complete clinical protection against challenge with the velogenic strain, boost with the genotypically-matched velogenic strain was far more effective at preventing virus shedding.  These findings confirm that the use of the local velogenic strain as boost vaccine is a more efficacious strategy to prevent Newcastle disease outbreaks.  In addition, the authors go on to show that, of the five known neutralizing epitopes on the NDV F protein relative to those on the lentogenic LaSota strain, only one exhibits a substitution, that being an A79T mutation the A3 antigenic site.  This could mean that efficacy of the booster vaccine strain depends to an extent on its homology with the challenge virus with a genotypic mismatch between vaccine and challenge virus resulting in a suboptimal protective response.

Response: Thank you for summarizing the study.

This is a very strong manuscript.  The experiments are all well designed and appropriately interpreted.  The data strongly support the use of a vaccination strategy that involves two priming vaccinations with a live lentogenic strain followed by a booster vaccination with a genotypically-matched inactivated local virus. There is only one minor criticism of the manuscript.  In the interests of completeness, arrows should be added to Figure 4 to indicate, wherever possible, the pathological lesions in panels a-d with an explanation added to the figure legend. 

Response: Arrows have been added to indicate the pathological lesions in the tissues.

Reviewer 4 Report

The manuscript is nicely done and written. The study design is appropriate and apparently, the analyses were carefully performed.  I believe that the results are valuable for the scientific community and has significant scientific merit, as it will probably ignite many further studies in the near future.

However, some minor points need to be clarified before the publication.

Line 51 – please expand the abbreviation of NDV as used for the first time.

Line 92 – please properly present your hypothesis.

Line 104 – please provide producer details of commercial LaSota vaccine.

Line 121 – please define city and the state code for Sigma-Aldrich.

Line 162 – WOAH acronym is used only one time. Therefore, there is no sense to abbreviate it.

Figure 4 – please add arrows for better presentation of the observed pathological changes.

Author Response

The manuscript is nicely done and written. The study design is appropriate and apparently, the analyses were carefully performed.  I believe that the results are valuable for the scientific community and have significant scientific merit, as it will probably ignite many further studies in the near future.

Response: Thank you for your appreciation.

However, some minor points need to be clarified before the publication.

Line 51 – please expand the abbreviation of NDV as used for the first time.

Response: Full form of the virus has been added

Line 92 – please properly present your hypothesis.

Response: The hypothesis that genotypic mismatch between the field virus and vaccine strain could potentially contribute to the vaccine failure in the field has been added (Line 104-106).

Line 104 – please provide producer details of commercial LaSota vaccine.

Response: The producer details has been added

Line 121 – please define city and the state code for Sigma-Aldrich.

Response: State code has been added.

Line 162 – WOAH acronym is used only one time. Therefore, there is no sense to abbreviate it.

Response: The abbreviation has been removed.

Figure 4 – please add arrows for better presentation of the observed pathological changes.

Response: Arrows have been added to indicated pathological changes in tissues.

Reviewer 5 Report

Hossain et al investigate protection from a velogenic NDV strain after different vaccination regimen. Specifically they compare a boost with live or killed lentogenic NDV (LaSota) (preceded by a double prime with live lentogenic NDV) to a boost with killed NDV C161/2010. Vaccinated animals then receive a challenge with live C161/2010 that is an exact match with the killed C161/2010 used in the booster. The most significant results are that all vaccinated birds were (pathologically) protected against challenge with a velogenic NDV field strain, and, more importantly perhaps, that a booster with killed C161/2010 virus almost completely prevented shedding. In general the work is straightforward, well written with well laid-out figures, and presents interesting results. The authors state that it was already reported that inactivated boosters may be better than live booster (ref 49), but provide little history/context to help the reader understand the rationale and any novelty of the present work and its chosen vaccination regimen (see below). It would also help to provide more explanation and discussion on the advantages and disadvantages of genotype-matched vaccines.

The following concerns were noted, which can mostly be addressed in the text.

Historic context is missing, making it difficult to understand the rationale of the vaccination strategies examined. For example, the need for genotype-matched NDV vaccines is not mentioned in the introduction. At the same time, there are drawbacks of genotype-specific vaccines, as NDV is constantly developing, and this also should be discussed. Another example: is the two-prime regimen an industry-standard? If so, it should be explained why. Also, do the authors agree that this is the best possible prime regimen, and did they consider a different priming method? A recommendation for this two-prime vaccination is made in the conclusion, but more context should be added as to explain this recommendation.

In methods, the hemagglutination test is referenced, but should be briefly described.

In Fig. 2C, what is the purpose of looking at antibody titers after challenge? Every repeat infection will surely affect the antibody titer, but the goal is to achieve a protective antibody titer prior to encountering a wildtype strain in the field.

Is the C161/2010 field strain a viscerotropic strain? Please add this information. Also add more infomation on which factors determine the outcome in terms of viscero vs neurotropic. In terms of cross-protection, it would have been nice to compare a viscero vs a neurotropic strain challenge. At the least, the authors should explain why only viscerotropic symptoms were encountered.

Table 3: needs more explanation. Have these epitopes been previously established? Are the epitopes known? Why is only 1 amino acid shown for four of the neutralizing epitopes whereas for A4, the entire epitope is shown.

A perfect match (between booster and challenge) would raise the expectation of better protection, and this is true at the level of virus shedding. However, will the improved shedding prevention hold up when the circulating strain is not a perfect match? If not, such a vaccine would have to be repeatedly adjusted based on newly emerging strains.

Authors believe that MDA interfered with first vaccine uptake. Although this is possible, it cannot be excluded that the first prime vaccination is poorly immunogenic. Has the live prime vaccination ever been done in birds without maternal antibodies? If maternal antibodies are a problem, is it known how long the maternally-derived protection lasts? If protection lasts a significant amount of time, then a prime on day 7 would be unnecessary, and the start of vaccination can be delayed to the point where maternal antibody levels go down.

Authors propose a re-focussing of the immune response could potentially have occurred based on Fig 2b,c. However, there is no significant difference between Lasota-killed and C161/2010 -killed boosters when C161/2010 HI antigen is used, and thus there is little evidence for re-focussing.

Author Response

Hossain et al investigate protection from a velogenic NDV strain after different vaccination regimen. Specifically, they compare a boost with live or killed lentogenic NDV (LaSota) (preceded by a double prime with live lentogenic NDV) to a boost with killed NDV C161/2010. Vaccinated animals then receive a challenge with live C161/2010 that is an exact match with the killed C161/2010 used in the booster. The most significant results are that all vaccinated birds were (pathologically) protected against challenge with a velogenic NDV field strain, and, more importantly perhaps, that a booster with killed C161/2010 virus almost completely prevented shedding. In general, the work is straightforward, well written with well laid-out figures, and presents interesting results.

Response: Thank you for summarizing the findings and appreciation.

The authors state that it was already reported that inactivated boosters may be better than live booster (ref 49) but provide little history/context to help the reader understand the rationale and any novelty of the present work and its chosen vaccination regimen (see below). It would also help to provide more explanation and discussion on the advantages and disadvantages of genotype-matched vaccines.

Response: The advantages and disadvantages of genotype-matched vaccines have been discussed (Line 421-429).

The following concerns were noted, which can mostly be addressed in the text.

Historic context is missing, making it difficult to understand the rationale of the vaccination strategies examined. For example, the need for genotype-matched NDV vaccines is not mentioned in the introduction.

Response: The need for genotype-matched vaccines has been addressed in the introduction (Line 83-92).

At the same time, there are drawbacks of genotype-specific vaccines, as NDV is constantly developing, and this also should be discussed.

Response: The limitation of genotype-matched vaccines and its potential solution has been discussed (Line 425-430).

Another example: is the two-prime regimen an industry-standard? If so, it should be explained why. Also, do the authors agree that this is the best possible prime regimen, and did they consider a different priming method? A recommendation for this two-prime vaccination is made in the conclusion, but more context should be added as to explain this recommendation.

Response: Most of the vaccine manufacturer recommend two priming regimens in ND endemic countries including Bangladesh. However, experimental data on the effectiveness of this regimen are limited which has been explored in this study. Our unpublished data using single priming showed only moderate clinical protection (about 80%) upon virulent NDV challenge but did not prevent virus shedding.

In methods, the hemagglutination test is referenced, but should be briefly described.

Response: The HI test method has been added (Line 180-187).

In Fig. 2C, what is the purpose of looking at antibody titers after challenge? Every repeat infection will surely affect the antibody titer, but the goal is to achieve a protective antibody titer prior to encountering a wildtype strain in the field.

Response: We completely agree with the reviewer. We included this figure as booster with LaSota (Live) vaccine mounted significantly lower antibody response compared to the inactivated LaSota and C161 vaccines (Figure 2b), but all vaccinated groups showed similar antibody responses upon challenge.

Is the C161/2010 field strain a viscerotropic strain? Please add this information. Also add more information on which factors determine the outcome in terms of viscero vs neurotropic. In terms of cross-protection, it would have been nice to compare a viscero vs a neurotropic strain challenge. At the least, the authors should explain why only viscerotropic symptoms were encountered.

Response: The BD-C161/2010 strain belongs to the velogenic viscerotropic pathotype of NDV (added in the materials and methods) and lesions resembling this viscerotropic pathotype were found in the control infected chickens in this study. Our previous sequential pathogenicity study of this strain showed profound neuropathogenicity of the virus in chickens (Kabiraj et al., 2020). The comparison between viscerotropic and neurotropic NDV in terms of cross-protection needs further study and beyond the scope of this study.

Table 3: needs more explanation. Have these epitopes been previously established? Are the epitopes known? Why is only 1 amino acid shown for four of the neutralizing epitopes whereas for A4, the entire epitope is shown.

Response: The five neutralizing epitopes have been established previously (Yusoff, 1989). Among them, four were discontinuous (A1-A3 and A5) and one linear (A4) epitope.

A perfect match (between booster and challenge) would raise the expectation of better protection, and this is true at the level of virus shedding. However, will the improved shedding prevention hold up when the circulating strain is not a perfect match? If not, such a vaccine would have to be repeatedly adjusted based on newly emerging strains.

Response: In countries, where more than one genotypes of virulent NDV are circulating, a multivalent inactivated vaccine could be formulated incorporating both field strains in the vaccine formulation (Discussed in the text line 428-430).

Authors believe that MDA interfered with first vaccine uptake. Although this is possible, it cannot be excluded that the first prime vaccination is poorly immunogenic. Has the live prime vaccination ever been done in birds without maternal antibodies? If maternal antibodies are a problem, is it known how long the maternally-derived protection lasts? If protection lasts a significant amount of time, then a prime on day 7 would be unnecessary, and the start of vaccination can be delayed to the point where maternal antibody levels go down.

Response: Live prime vaccination has been done in birds without material antibodies in studies which used SPF chicks (e.g., Wazid et al., 2018). Indeed, MDA is a challenge in successful immunization in NDV endemic countries when hens are routinely vaccinated against NDV. We also detected a relatively high MDV titer during the first week of age which could be protective. However, there are variations in the MDA titer among birds, which could lead to outbreaks in the unvaccinated flocks and the protective efficacy of MDA against virulent NDV challenge needs further study.

Authors propose a re-focusing of the immune response could potentially have occurred based on Fig 2b, c. However, there is no significant difference between Lasota-killed and C161/2010 -killed boosters when C161/2010 HI antigen is used, and thus there is little evidence for re-focusing.

Response: The sentence has been modified (line 397-398).

Round 2

Reviewer 1 Report

The study has been improved.